# STAndardised DIagnostic Assessment for children and young people with emotional difficulties (STADIA): protocol for a multicentre randomised controlled trial

Florence Day ,[1] Laura Wyatt,[1] Anupam Bhardwaj,[2] Bernadka Dubicka ,[3,4] Colleen Ewart,[5] Julia Gledhill,[6] Marilyn James,[1] Alexandra Lang ,[7] Tamsin Marshall,[8] Alan Montgomery,[1] Shirley Reynolds,[9] Kirsty Sprange ,[1] Louise Thomson,[10] Ellen Bradley,[10] James Lathe,[1] Kristina Newman,[11] Chris Partlett,[1] Kath Starr ,[1] Kapil Sayal [10,11]

FD and LW are joint first authors.

For numbered affiliations see end of article.

**Correspondence to**
Professor Kapil Sayal;
kapil.sayal@nottingham.ac.uk

## ABSTRACT

**Introduction** Emotional disorders (such as anxiety and depression) are associated with considerable distress and impairment in day-to-day function for affected children and young people and for their families. Effective evidence-based interventions are available but require appropriate identification of difficulties to enable timely access to services. Standardised diagnostic assessment (SDA) tools may aid in the detection of emotional disorders, but there is limited evidence on the utility of SDA tools in routine care and equipoise among professionals about their clinical value.

**Methods and analysis** A multicentre, two-arm, parallel group randomised controlled trial, with embedded qualitative and health economic components. Participants will be randomised in a 1:1 ratio to either the Development and Well-Being Assessment SDA tool as an adjunct to usual clinical care, or usual care only. A total of 1210 participants (children and young people referred to outpatient, specialist Child and Adolescent Mental Health Services with emotional difficulties and their parent/carers) will be recruited from at least 6 sites in England. The primary outcome is a clinician-made diagnosis about the presence of an emotional disorder within 12 months of randomisation. Secondary outcomes include referral acceptance, diagnosis and treatment of emotional disorders, symptoms of emotional difficulties and comorbid disorders and associated functional impairment.

**Ethics and dissemination** The study received favourable opinion from the South Birmingham Research Ethics Committee (Ref. 19/WM/0133). Results of this trial will be reported to the funder and published in full in the Health Technology Assessment (HTA) Journal series and also submitted for publication in a peer reviewed journal.

**Trial registration number** ISRCTN15748675; Pre-results.

## Strengths and limitations of this study

⇒ Large real-world multicentre randomised controlled trial of the Development and Well-Being Assessment (DAWBA) Standardised Diagnostic Assessment tool as an adjunct to usual care versus usual care only.

⇒ Trial procedures are carried out remotely with all data collection and the DAWBA completed online or via telephone, facilitating post-trial implementation into future service delivery models and routine clinical care.

⇒ The embedded health economic component permits evaluation of both clinical and cost-effectiveness.

⇒ Embedded qualitative work will support optimal delivery and implementation to enhance acceptability, effectiveness and long-term uptake.

⇒ Participants, researchers and clinicians cannot be blinded to treatment allocation.

people (CYP) and their families, with adverse effects on family and peer relationships, quality of life, social involvement and activities, academic attainment and occupational opportunities, ultimately affecting life chances.[1–4] Emotional disorders are frequently comorbid with other disorders,[2 5] and are associated with self-harm and completed suicide. Effective evidence-based interventions are available but require appropriate identification of presenting difficulties to enable timely access to services and earlier recovery.[3]

The prevalence of emotional disorders has increased considerably over the past two decades.[1] In the UK, CYP with clinically significant emotional difficulties may be referred to outpatient specialist Child and

## INTRODUCTION

Emotional disorders cause considerable distress for affected children and young

Adolescent Mental Health Services (CAMHS). However, insufficient information is a common reason for referrals being declined.[6] There is limited evidence to inform optimal approaches to determine which referrals should be accepted, contributing to a large variation in acceptance rates.[6] Likewise there is a lack of evidence on how best to conduct assessments for suspected emotional difficulties to optimise outcomes. Acceptance criteria and assessment procedures differ across services and there is no single standardised approach.

The multidisciplinary nature of CAMHS means CYP are assessed by practitioners from different professional backgrounds, with variations in training, ethos and conceptualisations of presenting difficulties. The type and scope of assessments offered vary. Assessments are often conducted by practitioners without formal diagnostic training[7] and recording of potential diagnostic information can be influenced by patient, clinician and service-related contextual considerations.[8] The validity and value of mental health diagnoses have been questioned, reflecting concerns around restricting service access,[9] stigma or labelling.[7 10 11] This can mean that in routine practice, assessments are often undertaken without the aim of making or recording a diagnosis.

However, National Institute for Health and Care Excellence (NICE) guidelines for management and treatment are usually based on diagnostic classification of disorders, so the ability to offer evidence-based interventions requires that the CYP's difficulties are appropriately identified. Although NICE Quality Standards[12] state that CYP with suspected depression should have the diagnosis confirmed and recorded, this is highly variable in practice.[7 13] The use of diagnostic assessments has been recommended so that important problems are detected and appropriate interventions are offered.[3 11] The NICE guidelines for depression have recommended the use of standardised diagnostic assessment (SDA) tools as potential adjuncts in the detection of depression within CAMHS.[14] It has further been recommended that SDA tools should be used as an adjunct to clinical assessments, potentially at the point of referral receipt, to enable the allocation of cases to the most appropriate professional.[10 15 16]

One such SDA tool is the Development and Well-Being Assessment (DAWBA), a structured package of questionnaires and interviews which can be completed online or by telephone and yields algorithm-based diagnostic information.[17] The DAWBA has established reliability and validity[17] and has been widely used for screening, diagnosis and outcome measurement in research in both clinical and community settings,[18 19] including trials of SDAs[20 21] and large scale epidemiological research.[1 22 23] A previous randomised controlled trial (RCT) using the DAWBA highlighted that, for emotional disorders, disclosing DAWBA diagnosis information to clinicians can improve the level of agreement between the DAWBA and clinical diagnoses, suggesting that the DAWBA can aid clinical detection of emotional disorders.[21] It also improved

detection of comorbid disorders. A UK trial found higher levels of agreement between DAWBA and clinical diagnoses, following disclosure of DAWBA information, in relation to anxiety disorders.[20] Practitioners acknowledged that the additional information could supplement the assessment and aid detection of difficulties.[10]

Hence, it might be expected that the introduction of an SDA tool following CAMHS referral receipt could enable resources to be better targeted and a timely conclusion to assessments with a diagnostic decision, increase the likelihood that an appropriate evidence-based treatment is offered, and lead to improved outcomes and better experience of care for CYP and their families. However, there is limited evidence on the utility of SDA tools for informing optimal approaches to assessment within routine clinical practice.

### Aims and objectives

The aim is to evaluate the clinical and cost-effectiveness of the DAWBA SDA tool, as an adjunct to usual clinical care for CYP presenting with emotional difficulties referred to CAMHS.

Specific objectives are to:

1. Conduct an RCT to determine the effectiveness of the DAWBA as an adjunct to usual clinical care on diagnosis and treatment of emotional disorders, symptoms of emotional difficulties and comorbid disorders and associated functional impairment.
2. Undertake an internal pilot to assess recruitment and acceptability.
3. Include a qualitative component within the pilot phase to address:
   a. The feasibility of recruitment.
   b. The acceptability and usability of the interventions and procedure.
   c. How the intervention is used and could be refined for the main trial.
4. Conduct a process evaluation alongside the main trial which will:
   a. Optimise the design and delivery of the DAWBA to enhance acceptability, effectiveness and long-term uptake.
   b. Identify the barriers and facilitators to implementation of the DAWBA from the perspectives of CYP, parents and CAMHS practitioners, managers and commissioners.
5. Estimate cost-effectiveness of the use of the DAWBA versus usual care.
6. Make evidence-based recommendations for assessment procedures within CAMHS and produce practice guidelines for clinical decision making around the referral acceptance and assessment processes.

### METHODS AND ANALYSIS
### Design

A multicentre, two-arm, parallel group RCT, with embedded qualitative and health economic components.

An internal pilot period, completed in the first 9 months of recruitment, will determine feasibility of recruitment and follow-up, assessed by the independent trial steering committee (TSC) against predefined stop/go criteria. The study start date is 1 November 2018 and end date is 31 October 2022.

## Setting

Recruitment will take place in at least six National Health Service (NHS) Trusts in England, providing outpatient multidisciplinary specialist CAMHS. Sites are geographically dispersed covering urban and rural areas, thus are likely to be sociodemographically representative of CAMHS referrals in England, enabling nationally generalisable findings.

## Recruitment and eligibility
### Participant identification

The population is CYP presenting with emotional difficulties referred to CAMHS. Participants are identified through the usual referral pathways for the participating sites, which includes NHS and local authority managed Single/Central Point of Access referral points as well as referrals directly received and processed by CAMHS teams.

The STADIA (STandardised DIagnostic Assessment for children and young people with emotional difficulties) trial researchers (NHS personnel, based within the CAMHS SPA/triage team to carry out research activities on behalf of the team and authorised to access referral information) at each site review the referrals received by CAMHS to identify CYP presenting with emotional difficulties, according to a standard proforma (online supplemental appendix 1). Referrals that mentioned any current emotional difficulties will be included, regardless of the number, frequency or severity of the emotional difficulties. Potentially eligible participants are invited to consider taking part in the trial and provided with written information. The initial invitation follows standardised wording to ensure clarity and consistency of approach.

Identification of participants takes place after referral receipt, but prior to referral acceptance (figure 1).

## Consent

Prior to consent, eligibility will be confirmed (box 1) during telephone contact with the local STADIA researcher, who will also provide written and verbal information about the trial, answer questions and support the electronic consent/assent process. Participants who are eligible and provide verbal consent to participation during the call will be provided with a personal link to the online electronic informed consent/assent form (table 1, online supplemental appendices 2 and 3, respectively), enabling them to provide written informed consent/assent.

The participation and consent/assent requirements for the trial are shown in table 1.

Participants are free to withdraw at any time and for any reason. Participants may withdraw from the intervention, follow-up questionnaires and/or data collection from records in any combination (eg, participants who do not complete the intervention will continue to be followed up, participants withdrawing from follow-up questionnaire completion may continue to consent for data collection from records). Withdrawn participants will not be replaced. Data collected prior to withdrawal will be retained and used in the analysis.

Where CYP aged 16 or 17 have consented for their own involvement they can continue to participate in the trial in the event of their parent/carer's withdrawal, however, the parent/carer involvement would not continue should the CYP withdraw consent.

## Randomisation and concealment

Participants will be randomised in a 1:1 ratio to either intervention or control. Allocation will be assigned using a minimisation algorithm balancing on recruiting site, CYP age (5–10, 11–15, 16–17 years) and sex, incorporating a probabilistic element to allocation. The allocation algorithm was created by Nottingham Clinical Trials Unit (NCTU) in accordance with their standard operating procedure. Allocation is concealed using an automated web system operated by NCTU.

Randomisation is automatically generated within the online system following submission, and automated verification, of baseline data by the primary participant. Participants are presented with their allocation and further instructions on-screen with email confirmation. Instructions for DAWBA completion are included for those in the intervention arm. Email confirmation is sent to the coordinating centre and site research team.

It will not be possible to blind participants, site researchers, clinicians and some trial staff to treatment allocation, but treatment allocation data will be restricted to those trial staff who require access to facilitate trial conduct. In particular, it will not be fully possible to blind researchers conducting data collection from records. However, any possible diagnoses identified from the CAMHS records will be recorded verbatim on the data capture form and will be subject to adjudication by the trial adjudication committee (members of the trial management group (TMG)). The committee will be blinded to treatment allocation and participant ID.

The risk of contamination between arms is considered low. Access to the DAWBA, and provision of the DAWBA report, is only provided to participants in the intervention arm. SDA tools are not current practice in standard care and it is unlikely that control participants will be asked to complete these at the point of referral receipt. DAWBA completion occurring outside the trial for control arm participants will be collected during follow-up.

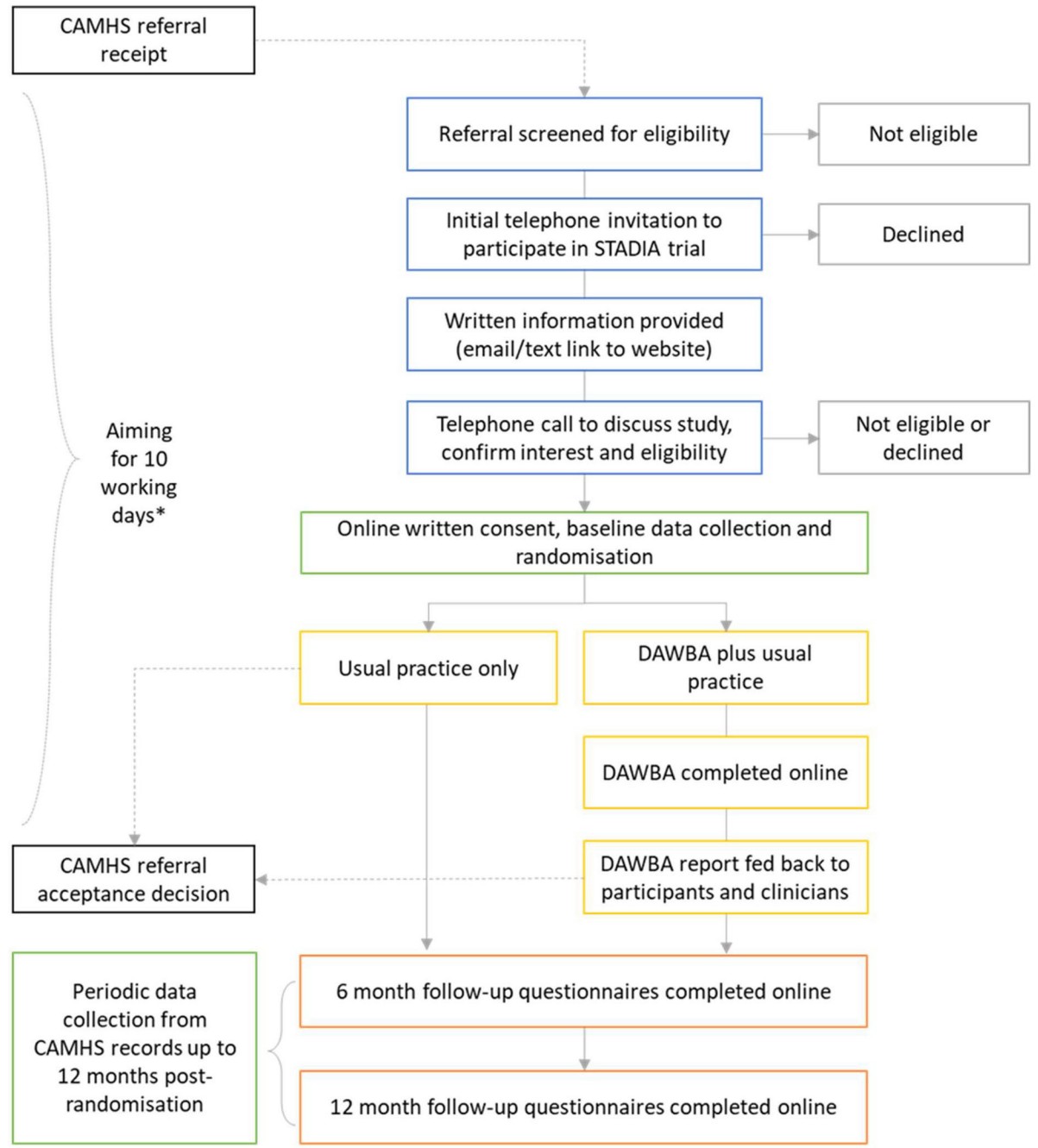

**Figure 1** Participant flow. *For sites where the waiting time for the CAMHS acceptance decision usually exceeds 10 working days from referral receipt, recruitment activities may start and/or continue beyond 10 working days from referral receipt, providing the intervention period can be completed prior to the CAMHS referral decision. CAMHS, Child and Adolescent Mental Health Services; DAWBA, Development and Well-Being Assessment. STADIA, STAndardised DIagnostic Assessment for children and adolescents with emotional difficulties.

## Interventions
### Development and Well-Being Assessment
The trial intervention is the DAWBA.[24] The DAWBA has a modular structure, with only those modules relevant to emotional and comorbid disorders included; separation anxiety, specific phobia, social phobia, panic and agoraphobia, generalised anxiety, post-traumatic stress disorder, obsessive–compulsive disorder, depression, oppositional defiant disorder and conduct disorder. Whereas, the strengths and difficulties questionnaire,

bipolar disorder and body dysmorphic disorder are not included in the STADIA-specific DAWBA report as these modules do not generate diagnostic predictions. No free-text responses are collected.

The DAWBA will be self-reported by participants via the secure, standalone online platform created and maintained by the DAWBA developer.[24] Access is by a unique ID number and password, assigned at the point of randomisation via a stock control system integrated into the randomisation system, ensuring accountability of DAWBA slot allocation.

## Box 1  Eligibility criteria

**Inclusion criteria for the children and young people (CYP)**
⇒ Aged 5–17 years.
⇒ Referred to outpatient multidisciplinary specialist Child and Adolescent Mental Health Services (CAMHS).
⇒ Presenting with emotional difficulties.
⇒ If aged <16, has an eligible individual with parental responsibility (see parent/carer eligibility criteria below) willing and able to participate in the trial.
⇒ If aged 16–17, has capacity to provide valid written informed consent.
⇒ If aged 16–17 and participating without a parent/carer, able to complete the assessment tool in English.
⇒ If aged 16–17 and participating without a parent/carer, access to internet and email or telephone.

**Exclusion criteria for the CYP**
⇒ Emergency or urgent referral to outpatient multidisciplinary specialist CAMHS (ie, requires an expedited assessment) according to local risk assessment procedures.
⇒ Severe learning disability.
⇒ Previously randomised in the STADIA trial.

**Inclusion criteria for the parent/carer**
⇒ Individual with parental responsibility for the CYP referred to CAMHS; this will be the CYP's mother or father, legally appointed guardian or a person with a residence order concerning the CYP.
⇒ Adequate knowledge of the CYP to be able to complete the assessment tool (ie, known for at least 6 months).
⇒ Has capacity to provide valid written informed consent.
⇒ Access to internet and email or telephone.
⇒ Able to complete the assessment tool in English.

**Exclusion criteria for the parent/carer**
⇒ Local authority representatives designated to care for the CYP.

The DAWBA may be completed by the parent/carer and/or CYP aged 11+, depending on the consent and participation arrangements (table 1) DAWBA completion will be monitored and the STADIA researcher will support and encourage completion. Participants will be able to complete the DAWBA in a telephone call with the STADIA researcher if required. Participants are asked to complete all modules of the DAWBA presented to them. Should the DAWBA be only partially completed by respondents the report will be based only on fully answered modules with missing responses identified as such.

A trial-specific DAWBA report will be prepared for each participant, based on a standard, study-specific template (online supplemental appendix 4). The algorithm-derived diagnostic predictions will be used to highlight the likelihood of a CYP meeting International Classification of Diseases (ICD-10) criteria for the disorders assessed; the report is based entirely on the algorithm-derived predictions and is not clinically rated. The report will be sent to participants (via post or email) and CAMHS clinicians (via upload to the clinical record), as an adjunct to usual clinical practice.

### Control

CYP randomised to the control arm will receive usual care (ie, referral review as usual). Based on standard information provided with the referral a clinical decision is made about whether the referral is accepted and, if so, a clinician conducts the initial CAMHS assessment as per usual practice in the service.

### Sample size

A target sample size of 1210 participants will be recruited and randomised, with equal allocation to intervention or control.

Assuming 45% of control participants have a confirmed diagnosis within 12 months (based on unpublished data obtained from the trial sites), detection of an absolute increase of 10% with 90% power and 5% two-sided alpha, requires 544 participants per arm for analysis. Allowing for up to 10% non-collection of the primary outcome, we will randomise 1210 participants.

### Measures and outcomes
#### Primary outcome

The primary outcome is a clinician-made diagnosis decision about the presence of an emotional disorder within 12 months of randomisation, that is, diagnosis of an emotional disorder will be coded as 'yes'; absence or uncertainty (for example, reflecting ongoing assessment or investigation) will be coded as 'no'. Eligible diagnoses are those that reflect 'emotional' or 'internalising' disorders in ICD/DSM (online supplemental appendix 5). The diagnosis must be documented in the clinical record within 12 months of randomisation by a mental health services clinician in an NHS-delivered or NHS-commissioned service.

Diagnoses will be collected from clinical records using a standard proforma. Alternative possible diagnoses identified from the clinical notes will be recorded verbatim on the data capture form and will be subject to adjudication by members of the TMG (online supplemental appendix 6).

#### Secondary outcomes

Secondary outcomes are detailed in table 2.

#### Health economic measures

Health-related quality of life (HRQoL) outcome measures are detailed in table 2.

#### Resource use

Data will be collected on healthcare, education and social care resource use for both the CYP and parents/carers, using a purposely designed resource use collection tool. The questionnaire was developed by the study's health economics team at Nottingham following discussions with the study's patient and public involvement (PPI) team and representatives. This was an iterative process until all parties including the PPI team and representatives, the health economics team and the wider TMG were reassured the questionnaire was fit for purpose. It

**Table 1** Consent and participation

| Age of CYP referred to CAMHS | CYP aged <11 | CYP aged 11–15 | | CYP aged 16–17 | |
|---|---|---|---|---|---|
| Initial contact with | Parent/carer | | | Depends on contact details provided with the CAMHS referral* | |
| Consent provided by | Parent/carer | Parent/carer | Parent/carer | CYP AND parent/carer (optional) | CYP |
| Assent provided by | None | CYP (optional) | None | None | None |
| Participant(s) | Parent/carer only | CYP and parent/carer dyad | Parent/carer only | CYP and parent/carer dyad | CYP only |
| Primary participant† | Parent/carer | Parent/carer | Parent/carer | CYP | CYP |
| Secondary participant | None | CYP | None | Parent/carer | None |
| DAWBA completed by | Parent/carer | Parent/carer AND CYP | Parent/carer | CYP AND parent/carer | CYP |
| Research questionnaires completed by | Parent/carer report on CYP Parent/carer self-report | Parent/carer report on CYP Parent/carer self-report CYP self-report | Parent/carer report on CYP Parent/carer self-report | CYP self-report Parent/carer report on CYP Parent/carer self-report | CYP self-report |

For all CYP aged <16 the initial contact about the study will be with the parent/carer. The involvement of CYP aged 11–15 will be at the discretion of the parent/carer.

*For CYP aged 16–17 if the CYP's contact details are provided on the CAMHS referral the first contact about the study will be with the CYP who can choose to nominate a parent/carer to participate in the trial alongside them or participate alone. If the parent/carer's contact details only are available the first contact will be with the parent/carer and the parent/carer will be asked whether the CYP can also be contacted but may choose to refuse this. The parent/carer will not be able to participate in the STADIA trial without the involvement or consent of the CYP.

†The primary participant is the person who must provide consent as a minimum requirement in order for randomisation to take place. Assent (of CYP aged 11–15) and parental consent (for CYP aged 16 and 17) may also be sought but is not mandatory and therefore will not be required prior to randomisation.

CAMHS, Child and Adolescent Mental Health Services; CYP, children and young people; DAWBA, Development and Well-Being Assessment; STADIA, STAndardised DIagnostic Assessment for children and adolescents with emotional difficulties.

collects data on all aspects of healthcare interventions including medication, inpatient and outpatient hospital visits and primary and community care use as well as societal and education costs. It also includes sections specifically designed to quantify the effect of time off work for parents/carers (including friends and family) to quantify the wider social cost, that is, implications for productivity. In addition, it measures effects on time lost from education or training for the child/young person because of emotional difficulties. A similar approach to capturing resource use information was employed by members of the study team for a feasibility trial involving parents and carers of children with attention-deficit/hyperactivity disorder (ADHD).[25]

These data will be attributable to the emotional difficulties of the young person and be self-reported by the parent/carer with online supplemental information obtained from CYP aged 16 and 17. Administrative records of treatments/interventions offered by CAMHS during the trial period may be considered as a online supplemental source of data.

## Sociodemographic data

The following sociodemographic data will be collected primarily from the participant-reported questionnaires; age at randomisation, sex, gender, ethnicity, paid employment and derived from the postcode of the child's primary residence, the index of Multiple Deprivation score.

## Data collection

Data will be collected through participant reported questionnaires (parent/carer and CYP self-report aged 11+) and from clinical records. Participant reported outcomes will be collected at baseline and 6 and 12 months post-randomisation (online supplemental appendix 7). Questionnaires are intended to be completed online by participants in the first instance—to maximise rates of completion and retention there will be an option for telephone completion, should participants have difficulty accessing or completing the questionnaires online.

Outcomes collected from records will be reported for the 12-month period following randomisation.

## Data management and analysis
### Data management

Arrangements for data handling are specified in the data management plan. Central and on-site monitoring will be carried out as required following a risk assessment and as documented in the monitoring plan. Monitoring activities will be carried out by the coordinating centre on behalf of the trial sponsor.

**Table 2** Secondary outcome definitions

| Outcome | Measurement | Definition |
|---|---|---|
| Acceptance of index referral | Collected from records | Whether the index referral (ie, the referral made to CAMHS at the point of recruitment to the STADIA trial) was accepted or declined. Acceptance is defined as being offered an appointment within CAMHS, whether or not the initial appointment was attended or subsequent appointments were offered/attended. Collected within 12 months of randomisation. |
| Acceptance of any referral within 12 months of randomisation | Collected from records | Whether the index referral or any subsequent referral to CAMHS (if made) was accepted or not. Acceptance as defined above for index referral. Collected within 12 months of randomisation. |
| Discharge from CAMHS within 12 months | Collected from records | Whether the child/young person was discharged from CAMHS (following acceptance of the index referral) during the 12 months post-randomisation. |
| Re-referral to CAMHS within 12 months | Collected from records | Whether the child/young person was re-referred to CAMHS (for those whose index referral was turned down by CAMHS or those whose index referral was accepted but were subsequently discharged) during the 12 months post-randomisation. |
| Confirmed diagnosis decision | Collected from records | Diagnosis of an emotional disorder or confirmed absence of an emotional disorder coded as 'yes' versus uncertainty about the presence of an emotional disorder coded as 'no'. Diagnosis as defined for primary outcome, collected within 12 months of randomisation. |
| Time from randomisation to diagnosis of emotional disorder | Collected from records | Date of diagnosis will be the first documented eligible diagnosis. Diagnosis as defined for primary outcome, collected within 12 months of randomisation. |
| Diagnoses made over the 12 month period from randomisation | Collected from records | The diagnosis must be documented in the clinical record within 12 months of randomisation by a mental health services clinician in an NHS-delivered or NHS-commissioned service. All diagnoses made within 12 months will be included. Measured using a standard proforma (pre-specified diagnoses). |
| Treatment offered for diagnosed emotional disorder | Collected from records | Whether treatment was offered for a diagnosed emotional disorder, as defined for primary outcome, collected within 12 months of randomisation. |
| Any treatment/ interventions given | Collected from records | All treatments/interventions offered by CAMHS for any reason within 12 months of randomisation, whether or not there is a documented diagnosis will be included. |
| Time from randomisation to the decision to offer treatment for a diagnosed emotional disorder | Collected from records | Date of decision will be the first date that the decision to offer treatment for a diagnosed emotional disorder is documented in the clinical notes, collected within 12 months of randomisation. |
| Time from randomisation to start of first treatment for a diagnosed emotional disorder | Collected from records | Date of treatment will be the first date that any treatment offered for a diagnosed emotional disorder is started. Treatment and diagnosed emotional disorder as defined, collected within 12 months of randomisation. |
| Time from randomisation to the decision to offer any treatment | Collected from records | Date of decision will be the first date that the decision to offer any treatment is documented in the clinical notes, collected within 12 months of randomisation. |
| Time from randomisation to start of any treatment | Collected from records | Date of treatment will be the first date that any treatment offered is started. Treatment as defined, collected within 12 months of randomisation. |
| Participant-reported diagnoses received in the 12 months post-randomisation | Participant self-report | Participants will be asked to report whether or not they received a diagnosis of the child/young person's difficulties from CAMHS in the 12 months post-randomisation and if so, what diagnosis was given and by whom. |
| Depression symptoms in the CYP | Mood and Feelings Questionnaire (MFQ) | MFQ[30] is a valid and reliable measure of depression in CYP.[31][32] 33 items are answered on a 3-point scale ('not true'=0, 'somewhat true'=1 point, 'true'=2 points). Scores range from 0 to 66 with higher scores indicating more severe depressive symptoms. A score of 27 or higher may be indicative of depression. MFQ collected at baseline, 6 and 12 months post-randomisation. |

Continued

**Table 2** Continued

| Outcome | Measurement | Definition |
|---------|-------------|------------|
| Anxiety symptoms in the CYP | Revised CYP's Anxiety Depression Scale (RCADS) | RCADS[33] is a 47-item questionnaire that measures the reported frequency of various symptoms of anxiety and low mood. Each item is rated on a 4-point scale (never=0, sometimes=1, often=2, always=3). An overall anxiety and low mood score is generated, with separate sub-scale scores for separation anxiety, social phobia, generalised anxiety, panic, obsessive–compulsive disorder and major depression. RCADS demonstrates good psychometric properties.[34] Total anxiety and depression scores range from 0 to 141. We will record scores for each of the six sub-scales. For analysis metric, we will use the total anxiety score. RCADS collected at baseline, 6 and 12 months post-randomisation. |
| Comorbid oppositional defiant/conduct disorder symptoms in the CYP | Strengths and Difficulties Questionnaire (SDQ) | SDQ[35]: A 25-item emotional and behavioural screening questionnaire for CYP. Each item is rated on a 3-point scale (not true, somewhat true, certainly true). Values of 0, 1 or 2 are assigned to each response. SDQ comprises five subscales and an impact supplement. The impact supplement asks effect of difficulties on homelife, friendships, education and leisure activities. SDQ has demonstrated reasonable psychometric properties.[36–39] Scores on the 'conduct problems' subscale will be used in the analysis of this outcome. Subscale scores range from 0 to 10. SDQ collected at baseline, 6 and 12 months post-randomisation. |
| Functional Impairment in the CYP | SDQ | Impact supplement scores will be used to determine functional impairment. Impact scores range from 0 to 10. Collected at baseline, 6 and 12 months postrandomisation. |
| Self-harm thoughts in the CYP | CYP self-report self-harm measure | CYP will be asked to report the frequency of thoughts of self-harm. Frequency of thoughts of self-harm are rated over the last 6 months in the following categories and scored accordingly: Not at all (0) Once or twice Three or more times Collected at baseline, 6 months and 12 months post-randomisation. |
| Self-harm behaviours in the CYP | CYP self-report self-harm measure | CYP will be asked to report frequency of instances of self-harm behaviour. Frequency of self-harm behaviour are rated over the last 6 months in the following categories and scored accordingly: Not at all (0) Once Two or more times Collected at baseline, 6 months and 12 months post-randomisation. |
| Depression symptoms in the parent/carer | Patient Health Questionnaire (PHQ-9) | PHQ-9:[40] PHQ-9 is frequently used as a screening tool for depression in general populations. Each of the nine DSM-IV depression criteria are scored as '0' (not at all) to '3' (nearly every day) depending on the frequency with which they were experienced over the last 2 weeks. Total scores range from 0 to 27 with higher scores indicating increased severity of depression, collected at baseline, 6 and 12 months post-randomisation. |
| Anxiety symptoms in the parent/carer | Generalised Anxiety Disorder Assessment (GAD-7) | GAD-7:[41] GAD-7 is a measure of the severity of anxiety in general populations. 7 items are rated according to the frequency with which they have been experienced over the past 2 weeks (0 = 'not at all', 1 = 'several days', 2 = 'more than half the days' and 3 = 'nearly every day'). Total scores range from 0 to 21 with higher scores indicating more severe anxiety. Collected at baseline, 6 and 12 months post-randomisation. |
| Time off education, employment or training because of emotional difficulties for the CYP | Resource use questionnaire | Days missed from education, employment or training (as applicable) for the CYP due to emotional difficulties. Collected at baseline, 6 and 12 months post-randomisation. |
| Health economic outcome measures | | |

Continued

**Table 2** Continued

| Outcome | Measurement | Definition |
|---|---|---|
| Health-related quality of life in the CYP | Child Health Utility 9D (CHU9D) and EuroQol Quality of Life Questionnaire 5 Domains for Young People (EQ-5D-Y) | CHU9D[42] consists of nine individual items with five levels of response per question (scored 1–5), that assess the CYP functioning 'today'. The following domains are included; worry, sadness, pain, tiredness, annoyance, school, sleep, daily routine and activities.<br><br>EuroQol-5D youth descriptive system[43] comprises five domains; mobility, looking after myself, doing usual activities, having pain or discomfort and feeling worried, sad or unhappy, values of 1, 2 or 3 are assigned to each response. The EuroQol Visual Analogue Scale (EQ-VAS) asks recipients to self-assess their health state 'today' from 0 (worst imaginable health) to 100 (best imaginable health), representing individual preferences. These measures will be self-reported by CYP aged 11+, with proxy versions also completed by the parent/carer for CYP <16.<br>Both collected at baseline, 6 and 12 months post-randomisation. |
| Health-related quality of life in the parent/carer | EuroQol Quality of Life Questionnaire 5 Domains, 5 Levels (EQ-5D-5L) | The EuroQol 5-dimension multi attribute utility instrument[44] comprises five domains; mobility, self-care, usual activities, pain/discomfort and anxiety/depression. Each domain is scored between 1 and 5. This descriptive profile, in combination with a valuation set, produces a single index for health status representing societal preferences. The index score ranges from –0.59 to 1, with 0 representing death, 1 of-perfect health and <0 of health states worse than death. The EQ-VAS is again included within the EQ-5D instrument Collected at baseline, 6 and 12 months post-randomisation. |

CAMHS, Child and Adolescent Mental Health Services; CYP, children and young people; NHS, National Health Service; STADIA, STAndardised DIagnostic Assessment for children and adolescents with emotional difficulties.

Data will be held on servers located within The University of Nottingham data centres. Security is both physical (secure limited access) and electronic (behind firewalls, access via user accounts). Personal data recorded on all documents will be regarded as strictly confidential and handled and stored in accordance with the Data Protection Act 2018.

## Statistical analysis

The primary approach to between-group comparative analyses will be by modified intention to treat (ITT) (ie, including all participants who have been randomised and without imputation of missing outcome data).

The primary comparative analysis will employ a generalised linear mixed model to compare the proportions in each group with a clinician-made diagnosis decision within 12 months of randomisation, adjusted for minimisation variables. The comparison will be presented as both an absolute (risk difference) and relative (risk ratio) effect, along with 95% CIs.

Secondary outcomes will be analysed using appropriate mixed effect regression models dependent on data type and will adjust for factors used in the minimisation and baseline value of the outcome where measured. For outcomes measured at multiple time points, these will be analysed using a mixed model with a treatment by time interaction to obtain estimates of treatment effect at each follow-up time.

Appropriate interaction terms will be included in the primary regression analyses in order to conduct subgroup analyses according to sex and age of the CYP.

Statistical analysis will be conducted using Stata V.17.0 (or later).

## Health economic analysis

In accordance with NICE guidance, primary analysis will take an NHS and personal social services perspective. Unit costs will be attached to participant reports of healthcare resource use or recorded treatments/interventions offered by CAMHS. The cost of the DAWBA itself will be distributed at the participant level across the intervention arm of the trial. Sensitivity analyses will take a wider perspective to capture the broader societal costs inclusive of out-of-pocket expenses and productivity losses. Indices of HRQoL for the EuroQol-5 Domains (EQ-5D), EuroQol-5D-Youth (EQ-5D-Y) and Child Health Utility instrument (CHU9D) will be derived using relevant population tariffs, and quality-adjusted life-years (QALY) estimated using area under the curve.

The economic evaluation will take an incremental approach between the two groups using an ITT population (irrespective of treatment received) and a 12-month time horizon. The outcome for the primary cost utility analysis will be the joint young person and parent/carer QALYs. The outcome for the secondary cost-effectiveness analysis will be confirmed diagnosis decisions. Outcomes will be paired with their respective direct-to-NHS costs, bootstrapped and scattered on the cost-effectiveness plane to characterise the uncertainty in incremental estimates. Using the net monetary benefit framework,[26] cost-effectiveness acceptability curves will be constructed to show the non-parametric probability the intervention is a cost-effective option, compared with usual care, across a range of willingness to pay thresholds per QALY, and within the secondary analysis per confirmed diagnosis decision. While the receipt of any diagnosis of emotional

difficulties in young people would likely lead to large divergences in lifecourse outcomes, the heterogeneity of conditions considered for diagnosis (online supplemental appendix 5) renders CUA modelling across the lifecourse infeasible. Secondary analysis is expected to be fully captured within the 12-month time horizon.

A full statistical analysis plan and health economics analysis plan will be developed and agreed prior to database lock and unblinding of the analysing statistician and health economist.

## Embedded qualitative study

During the internal pilot, semi-structured interviews are undertaken with a sample of participants who consented to be invited to participate in qualitative interviews. Researchers, clinicians, service managers and commissioners are identified by site leads. The proposed sample size is 25 participants (parent/carer and CYP aged 16–17), 25 staff and 15 service managers and commissioners. Interviews address: (1) the feasibility of recruitment; (2) the acceptability and usability of the interventions and procedure; (3) how the intervention is used and how this deployment could be refined for the main trial. Interviews are conducted by the qualitative researcher (KN) in person, or by phone or video call based on participant preferences and pandemic restrictions.

A process evaluation, conducted during the main trial phase, will aim to identify the barriers and facilitators to implementation of the intervention. Semi-structured interviews will be conducted with a further sample of participants and clinicians to explore the perceived functioning of the intervention, the organisation of the service and reflective experiences on outcomes.

Qualitative interview data will be recorded and encrypted on a password-protected Dictaphone and transferred securely to medical transcription company Dict8 for transcription. Transcriptions will be anonymised. Audio files will be destroyed after transcripts have been checked. Anonymised transcriptions will be analysed and stored on password protected computers and the secure University of Nottingham server.

## Qualitative analysis

All qualitative interview data will be initially analysed by the qualitative researcher (KN) using interpretative thematic approaches to coding, and adopt the framework method,[27] with input from the qualitative lead (LT), chief investigator (KSa) and PPI leads (CE and AL). NVivo V.12 will be used to manage the qualitative data.

## Patient and public involvement

Prior to submission, the proposal was informed by consultations with a person with lived parent/carer experience of CAMHS, including contribution to and review of the proposal, recruitment strategy, participant trial experience and consideration of burden of the intervention, and establishing a PPI workstream.

Following award, the PPI Co-Investigator team recruited two representatives naïve of the study design to provide independent review of the trial via their membership of the TSC. Both TSC members are persons with lived parent/carer experience of CAMHS.

During study set up, PPI Co-Investigator expertise was used to support researcher recruitment via the design and deployment of role plays within interviews.[28] This was to gain insight into candidates' capabilities when dealing with sensitive and challenging participant scenarios. Additionally, they contributed to design of researcher training materials, to support standardised approaches across trial sites. Iterative and creative design PPI activities were integral in the development of the STADIA trial logo and branding to ensure accessibility and acceptability to CYP and parents.

Since study commencement participatory design approaches have seen PPI co-design of the resource use questionnaire, qualitative interviews and the protocol for a Study Within A Trial to support participant engagement with follow-up. Additionally, collaborative working between the PPI and Qualitative workstreams has enabled examination of the qualitative themes using principles of the Framework Method[27] for independent verification of those themes.

Two PPI advisory panels have been established, meeting on average every 3 months since month 9 of the study. 'STADIA PPI Panel' has eight adult members, with lived parent/carer experience of CAMHS. 'STADIA Labs' has six CYP members, aged 15–19 at inception, with lived experience of CAMHS. These groups have been involved in many traditional activities such as review of the Participant Information Sheet (PIS) and consent forms, consultation on language and content for participant reminder text messages. PPI coproduction activities are also seeing the development of age appropriate study newsletters and the design of STADIA information videos including decision making about video concept, audience, message, aesthetic and content. PPI group members are provided with supplementary training about PPI practices and involvement opportunities. Due to the COVID-19 pandemic, PPI meetings have had to move online and so the PPI team are investing in knowledge transfer and upskilling PPI representatives in different ways of working and collaborating online.

There are a range of planned flexible opportunities for participating in project feedback and dissemination activities including cofacilitating and presenting at the interactive dissemination workshop/consensus meeting, publication authorship as peer researcher and presenting at conferences to showcase the project findings.

## ETHICS AND DISSEMINATION
### Ethics

The study was reviewed and received favourable opinion from the South Birmingham Research Ethics Committee (Ref. 19/WM/0133) on 12 June 2019; subsequent

amendments have been approved. The current, approved protocol is version 4.0 dated 3 February 2021.

## Safety

The trial intervention is conceptually similar to usual clinical practice (ie, CYP referred to CAMHS may be sent questionnaires about their difficulties), therefore, the risks of the trial are considered comparable. The DAWBA is widely used in research for data collection therefore, although used as an intervention in the STADIA trial, the risks may be regarded as similar to those of an observational/questionnaire study. Data to inform safety oversight will therefore be collected during routine follow-up, from existing outcome measures. There is no separate adverse event or serious adverse event reporting.

The number of participants meeting predefined safety outcomes will be reported on an ongoing basis to the TMG and TSC. Data will be presented by arms to the data monitoring committee (DMC).

## Trial oversight

Nottinghamshire Healthcare NHS Foundation Trust will undertake role of Sponsor as defined by the UK Policy Framework for Health and Social Care Research.[29] Delegated responsibilities will be assigned to the chief investigator, participating NHS Trusts and the trial coordinating centre, NCTU.

The full co-applicant team and NCTU staff responsible for the day-to-day management of the trial will form the TMG, responsible for monitoring recruitment and retention rates and implementing strategies to ensure targets are met. Independent trial steering and DMC will operate in accordance with trial-specific Charters.

## Dissemination

Results of this trial will be reported to the funder and published in full in the Health Technology Assessment (HTA) Journal series and also submitted for publication in a peer-reviewed journal.

**Author affiliations**
[1]Nottingham Clinical Trials Unit, School of Medicine, University of Nottingham, Nottingham, UK
[2]Cambridgeshire and Peterborough NHS Foundation Trust, Fulbourn, UK
[3]Division of Neuroscience and Experimental Psychology, University of Manchester, Manchester, UK
[4]Pennine Care NHS Foundation Trust, Ashton-under-Lyne, UK
[5]STADIA Patient and Public Involvement co-lead, Institute of Mental Health, University of Nottingham, Nottingham, UK
[6]Central and North West London NHS Foundation Trust, London, UK
[7]Faculty of Engineering, University of Nottingham, Nottingham, UK
[8]Berkshire Healthcare NHS Foundation Trust, Bracknell, UK
[9]School of Psychology and Clinical Language Sciences, University of Reading, Reading, UK
[10]Unit of Mental Health & Clinical Neurosciences, School of Medicine, University of Nottingham, Nottingham, UK
[11]Institute of Mental Health, Nottinghamshire Healthcare NHS Foundation Trust, Nottingham, UK

**Acknowledgements** We would like to acknowledge and thank all the parent/carers and young people participating in the trial and the research sites involved in recruiting participants and data collection. The authors would also like to thank the wider STADIA team for their input, including the PPI Advisory Panels, members of the independent Trial Steering and Data Monitoring Committees, and the Nottingham Clinical Trials Unit, who are the trial coordinating centre. Finally, thanks to the trial sponsor, Nottinghamshire Healthcare NHS Foundation Trust ( researchsponsor@nottshc.nhs.uk).

**Contributors** FD, LW, AB, BD, CE, JG, MJ, AL, TM, AM, SR, KSp, LT, EB, JL, KN, CP, KSt and KSa made substantial contributions to conception and design or acquisition of data; took part in drafting the article or revising it critically for important intellectual content; agreed to submit to the current journal; gave final approval of the version to be published; and agree to be accountable for all aspects of the work. KSa is guarantor for the paper. FD and LW contributed equally to this paper.

**Funding** This study was funded as a result of a commissioned call by the National Institute for Health Research (NIHR) Health Technology Assessment programme (Grant Reference Number 16/96/09).

**Disclaimer** The views expressed are those of the authors and not necessarily those of the NIHR or the Department of Health and Social Care. The funder will have no role in the collection, management, analysis, and interpretation of data; writing of the report; and the decision to submit the report for publication.

**Competing interests** None declared.

**Patient and public involvement** Patients and/or the public were involved in the design, or conduct, or reporting, or dissemination plans of this research. Refer to the Methods section for further details.

**Patient consent for publication** Not applicable.

**Provenance and peer review** Not commissioned; externally peer reviewed.

**ORCID iDs**
Florence Day http://orcid.org/0000-0003-0306-5558
Bernadka Dubicka http://orcid.org/0000-0002-8907-8589
Alexandra Lang http://orcid.org/0000-0002-7332-9443
Kirsty Sprange http://orcid.org/0000-0001-6443-7242
Kath Starr http://orcid.org/0000-0003-3356-7751
Kapil Sayal http://orcid.org/0000-0002-2050-4316

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
