## [Reviewer comments · BMJ Open]

ARTICLE DETAILS

TITLE (PROVISIONAL)	STANDARDISED Diagnostic Assessment for children and young people with emotional difficulties (STADIA): protocol for a multi-centre randomised controlled trial
AUTHORS	Day, Florence; Wyatt, Laura; Bhardwaj, Anupam; Dubicka, Bernadka; Ewart, Colleen; Gledhill, Julia; James, Marilyn; Lang, Alexandra; Marshall, Tamsin; Montgomery, Alan; Reynolds, Shirley; Sprange, Kirsty; Thomson, Louise; Bradley, Ellen; Lathe, James; Newman, Kristina; Partlett, Chris; Starr, Kath; Sayal, Kapil

VERSION 1 – REVIEW

REVIEWER	Ramchandani, Paul University of Cambridge, Education
REVIEW RETURNED	04-Dec-2021

GENERAL COMMENTS	Thank you for asking me to review this protocol for an interesting study. The manuscript is very clearly written and presented. It is especially good to see such a clear description of PPI engagement in the study – seems comprehensive. I have the following queries comments. 1. The abstract summarises the key aspects of the study succinctly and accurately. However, there is one area where greater clarity would be helpful is in the definition of outcome. In the abstract it states: “The primary outcome is diagnosis of an emotional disorder within 12-months post-randomisation.” But in the text at page 11 it states “The primary outcome is a clinician-made diagnosis decision about the presence of an emotional disorder within 12 months of randomisation.” The second could include a decision about absence of a disorder, but the first would not. 2. In recruitment section it states that one criteria is presenting with emotional difficulties – how is this operationalised? Although the symptoms are listed in appendix 1 it would be good to have some examples given in the main text along with information about how inclusion is determined – is it number of symptoms, or any symptoms, severity criteria, or something else? 3. The primary outcome is a diagnostic decision having been made (although see point 1 above). Although this is not critical, it would be interesting to see if there is agreement between the diagnosis made and the DAWBA indicative diagnosis – although I appreciate that this goes beyond the core question of the study, it would be useful to know if the intervention prompts an accurate diagnostic decision, rather than
---

	just prompting a decision. I appreciate this would only be available for the intervention arm. 4. Is collection of data from CAMHS records conducted blind to allocation? It is not completely clear from the description that this will always be the case. Is there a standard method for extracting details of diagnostic decisions.
--	---

REVIEWER	Montoya-Castilla, Inmaculada Universitat de Valencia Facultat de Psicologia, Personality, Assessment and Psychological Treatments
REVIEW RETURNED	03-Jan-2022

GENERAL COMMENTS	STANDARDISED Diagnostic Assessment for children and young people with emotional difficulties (STADIA): protocol for a multi-centre randomised controlled trial The manuscript entitled "STANDARDISED Diagnostic Assessment for children and young people with emotional difficulties (STADIA): protocol for a multi-centre randomised controlled trial" is considered relevant for research in the field of emotional disorders in childhood and adolescence. This is a protocol study that will include 1,210 adolescents aged between 5 and 17 years. The main purpose of the study is to evaluate the clinical and cost effectiveness of the DAWBA SDA tool, as an adjunct to usual clinical care for children and young people presenting with emotional difficulties referred to CAMHS. In order to improve some of the sections of the manuscript, some comments are made that could be useful: Introduction a) To make it easier to understand the acronym (CYP), it would be appreciated if "children and young people" were capitalised (line 6, page 3). b) On page 3 (line 10), the authors state that "Emotional disorders are frequently comorbid with other disorders [2, 5], and are associated with self-harm and completed suicide". Do studies (references 2 and 5) also support that emotional disorders are associated with self-harm and completed suicide? Methods and analysis a) Table 1 is not referenced in the text. Following the suggestions of the guidelines, authors are advised to mention it before it appears in the manuscript. Recruitment and eligibility a) The age inclusion criterion is children and adolescents aged 5-17 years. According to the World Health Organisation, adolescence extends to 19 years of age. Could the authors explain why 18 and 19 year olds will be excluded? Consent
--

	a) In Table 2, authors are requested to eliminate the use of capital letters in the questions and to use a heading that facilitates the understanding of the table. Interventions a) The content of Table 3 is considered to be succinct enough to avoid the use of a table. Authors are requested to mention in the text the included and excluded modules grouped together instead of using a table. Measures and outcomes. a) Will the characteristics of the diagnosing clinicians (e.g. years of experience, training, etc.) be taken into account? If this is not taken into account, a possible limitation of the study will have to be considered. b) Table 4 does not provide sufficient information about the assessment instruments to be used in the study. It lacks information about the psychometric properties of the questionnaires, validations in the adolescent population, the number of items, the type of scale and for which ages they are validated, among others. It is recommended that Table 4 be completed with the information in Appendix 5 c) Would it be possible for the authors to indicate which socio-demographic data will be collected? d) In Health economic measures section, the authors state that “The questionnaire was developed by health economists, in tandem with feedback from PPI representatives, addressing primary, secondary, and social care costs, alongside the broader patient-borne costs”. Would it be possible for the authors to explain the questionnaire in more detail, and could they mention any research in which it has been used with adolescents? e) In the abstract the authors state that the study was registered on 29 May 2019. Could the authors please detail what steps have been taken during the almost two years until 2021? It would be appreciated if the authors could include a time schedule for the study section. How long will the study last? What will be the procedure and the tasks they will perform each month? Data management and analysis a) Could the authors specify which software will be used for the statistical analyses? In the case of qualitative analysis, they mentioned NVIVO, but did not indicate which ones will be used for quantitative analysis. References a) The reviewers would appreciate a review of the current literature, especially in relation to the theoretical introduction and the studies conducted. More than 85% of the references, which the manuscript is based on, are older than 5 years and there is only one "current" reference from 2019 (which is two years old). Authors are kindly requested to update the references.
--	---

	b) A formal revision of the references is suggested. Following the line of the journal, it is recommended to use the Vancouver citation style. Other comments a) The warning “Error! Reference source not found” appears in numerous sections of the manuscript (page 5 line 29; page 8 line 39, page 9 lines 19 and 25; page 10 line 40; page 11 line 48). The authors are kindly requested to check the error.
--	---

VERSION 1 – AUTHOR RESPONSE

Reviewer 1 comments

Thank you for asking me to review this protocol for an interesting study. The manuscript is very clearly written and presented. It is especially good to see such a clear description of PPI engagement in the study – seems comprehensive.

Response: Thank you for your kind comments.

Comment 5: The abstract summarises the key aspects of the study succinctly and accurately. However, there is one area where greater clarity would be helpful is in the definition of outcome. In the abstract it states: “The primary outcome is diagnosis of an emotional disorder within 12-months post-randomisation.” But in the text at page 11 it states “The primary outcome is a clinician-made diagnosis decision about the presence of an emotional disorder within 12 months of randomisation. The second could include a decision about absence of a disorder, but the first would not.

Response: The abstract has word-count restrictions but we agree that, on reflection, the abstract wording should have made clear that the primary outcome is the presence of a diagnosis, according to the clinician i.e. diagnosis of an emotional disorder is coded as ‘yes’; absence or uncertainty (for example, reflecting ongoing assessment / investigation) about the presence of an emotional disorder is coded as ‘no’. We have expanded the abstract wording to clarify this and maintain consistency with the text on page 11.

Comment 6: In recruitment section it states that one criteria is presenting with emotional difficulties – how is this operationalised? Although the symptoms are listed in appendix 1 it would be good to have some examples given in the main text along with information about how inclusion is determined – is it number of symptoms, or any symptoms, severity criteria, or something else?

Response: The following text has been added to clarify screening processes “Referrals that mentioned any current emotional difficulties will be included, regardless of the number, frequency or severity of the emotional difficulties” (page 6 lines 207-209).

Comment 7: The primary outcome is a diagnostic decision having been made (although see point 1 above). Although this is not critical, it would be interesting to see if there is agreement between the diagnosis made and the DAWBA indicative diagnosis – although I appreciate that this goes beyond the core question of the study, it would be useful to know if the intervention prompts an accurate diagnostic decision, rather than just prompting a decision. I appreciate this would only be available for the intervention arm.

Response: We thank the reviewer for this interesting suggestion to further explore the level of concordance between the DAWBA and clinician-made diagnoses. As the reviewer highlights this would not form a comparative analysis between the intervention groups and so is beyond the scope of this protocol. However, we acknowledge and welcome the reviewer's suggestion that this would make for an interesting post-hoc exploratory analysis.

Comment 8: Is collection of data from CAMHS records conducted blind to allocation? It is not completely clear from the description that this will always be the case. Is there a standard method for extracting details of diagnostic decisions.

Response: While researchers responsible for collecting data from CAMHS records will not be systematically unblinded to randomised treatment allocation it is possible that they could become unblinded during the data collection process. Researchers are asked to record their opinion of treatment allocation before the record review to ascertain the extent of prior unblinding (e.g. through contact with participants). The content of the CAMHS clinical record (e.g. presence or otherwise of a DAWBA) will potentially unblind the researcher. This has been clarified in the blinding section (page 8, lines 257-260).

However, any possible diagnoses identified from the CAMHS records will be recorded verbatim on the data capture form and will be subject to adjudication by members of the Trial Management Group. The trial adjudication committee will be blinded to participant ID and to treatment allocation. The following text has been added "However, any possible diagnoses identified from the CAMHS records will be recorded verbatim on the data capture form and will be subject to adjudication by the trial adjudication committee (members of the Trial Management Group). The committee will be blinded to treatment allocation and participant ID." (page 8, lines 260-263)

Reviewer 2 comments

The manuscript entitled "STANDARDISED Diagnostic Assessment for children and young people with emotional difficulties (STADIA): protocol for a multi-centre randomised controlled trial" is considered relevant for research in the field of emotional disorders in childhood and adolescence. This is a protocol study that will include 1,210 adolescents aged between 5 and 17 years. The main purpose of the study is to evaluate the clinical and cost effectiveness of the DAWBA SDA tool, as an adjunct to usual clinical care for children and young people presenting with emotional difficulties referred to CAMHS. In order to improve some of the sections of the manuscript, some comments are made that could be useful:

Introduction:

Comment 9: (Introduction) To make it easier to understand the acronym (CYP), it would be appreciated if "children and young people" were capitalised (line 6, page 3).

Response: Agreed, we have capitalised "Children and Young People".

Comment 10: On page 3 (line 10), the authors state that "Emotional disorders are frequently comorbid with other disorders [2, 5], and are associated with self-harm and completed suicide". Do studies (references 2 and 5) also support that emotional disorders are associated with self harm and completed suicide?

Response: These papers did not investigate whether emotional disorders are associated with self-harm and completed suicide.

Methods and analysis

Comment 11: Table 1 is not referenced in the text. Following the suggestions of the guidelines, authors are advised to mention it before it appears in the manuscript.

Response: Table 1 is referenced in the Consent section of the methods “Prior to consent, eligibility will be confirmed (table 1)” (page 6, line 217).

Recruitment and eligibility

Comment 12: The age inclusion criterion is children and adolescents aged 5-17 years. According to the World Health Organisation, adolescence extends to 19 years of age. Could the authors explain why 18 and 19 year olds will be excluded?

Response: Thank you for your comment. Young people up to 17 years are included to reflect the upper age of the adolescents referred to and seen by Child and Adolescent Mental Health Services (CAMHS) for specialist help. At 18 years old entry is into adult mental health services. As per the NIHR-commissioned funding call (“Standardised diagnostic assessment tool as an adjunct to clinical practice in child and adolescent mental health services”) that led to this trial, we are focusing on referrals to CAMHS only. We acknowledge that some YP aged 17 on study entry will turn 18 years old within the 12-months following randomisation. STADIA researchers collecting outcome data from records are asked to access records from the adult mental health service record of the participant if applicable. Our primary outcome must be documented in the clinical record within 12 months of randomisation by a mental health services clinician in an NHS-delivered or NHS-commissioned service, which covers both CAMHS and adult mental health services.

Consent

Comment 13: In Table 2, authors are requested to eliminate the use of capital letters in the questions and to use a heading that facilitates the understanding of the table.

Response: Table 2 has been updated as per the request.

Interventions

Comment 14: The content of Table 3 is considered to be succinct enough to avoid the use of a table. Authors are requested to mention in the text the included and excluded modules grouped together instead of using a table.

Response: The content of Table 3 has been summarised in the text as follows; ...included; separation anxiety, specific phobia, social phobia, panic and agoraphobia, generalised anxiety, post-traumatic stress disorder (PTSD), obsessive compulsive disorder (OCD), depression, oppositional defiant disorder (ODD) and conduct disorder. Whereas, the strengths and difficulties questionnaire, bipolar disorder, and body dysmorphic disorder are not included in the STADIA-specific DAWBA report as these modules do not generate diagnostic predictions” (page 9, lines 275-280).

Measures and outcomes.

Comment 15: Will the characteristics of the diagnosing clinicians (e.g. years of experience, training, etc.) be taken into account? If this is not taken into account, a possible limitation of the study will have to be considered.

Response: In our case record review we are capturing the name and clinical role of the diagnosing clinician however we are not systematically recording their characteristics, level of training or experience. STADIA is a pragmatic trial, and the effectiveness of the DAWBA in a real-world clinical setting is being evaluated. A diverse and multi-disciplinary range of CAMHS health-care professionals will be involved in diagnosing emotional disorders in the children/young people. We do

not believe this represents a major limitation since randomisation should ensure balance with respect to diagnosing-clinician across intervention arm.

Comment 16: Table 4 does not provide sufficient information about the assessment instruments to be used in the study. It lacks information about the psychometric properties of the questionnaires, validations in the adolescent population, the number of items, the type of scale and for which ages they are validated, among others. It is recommended that Table 4 be completed with the information in Appendix 5.

Response: Thank you for your suggestion. We have replaced original table 4 with the information presented in Appendix 5 (new table 3, page 11) and supplemented with additional information.

Comment 17: Would it be possible for the authors to indicate which socio-demographic data will be collected?

Response: The following socio-demographic data are collected primarily from the participant-reported questionnaires: age at randomisation, sex, gender, ethnicity, education, paid employment, and, derived from the postcode of the child's primary residence the index of Multiple Deprivation score. This text has been added into the manuscript (page 18, lines 368-371).

Comment 18 In Health economic measures section, the authors state that "The questionnaire was developed by health economists, in tandem with feedback from PPI representatives, addressing primary, secondary, and social care costs, alongside the broader patient-borne costs". Would it be possible for the authors to explain the questionnaire in more detail, and could they mention any research in which it has been used with adolescents?

Response: This questionnaire collects participant-reported information about healthcare, education, and social care resource use for both the CYP and parents/carers. For example, it collects data on all aspects of healthcare interventions including medication, inpatient and outpatient hospital visits and primary and community care use as well as societal and education costs. It also includes sections specifically designed to quantify the effect on time off work for parents/carers (including friends and family) and the implications for productivity. In addition, it will seek to measure effects on time lost from education or training for the child/young person because of emotional difficulties.

A similar approach to capturing resource use information was employed by members of the study team for a feasibility trial involving parents and carers of children with ADHD (Hall et al. 2018. Protocol investigating the clinical utility of an objective measure of attention, impulsivity and activity (QbTest) for optimising medication management in children and young people with ADHD 'QbTest Utility for Optimising Treatment in ADHD' (QUOTA): a feasibility randomised controlled trial BMJ Open).

Please note that this additional information has not been added to the manuscript due to exceeding word count limitations. If the editor requires, we are happy to add this in.

Comment 19: In the abstract the authors state that the study was registered on 29 May 2019. Could the authors please detail what steps have been taken during the almost two years until 2021? It would be appreciated if the authors could include a time schedule for the study section. How long will the study last? What will be the procedure and the tasks they will perform each month?

Response: As requested by the Editor, the study start and end dates have now been included in the methods section. The first site opened to recruitment, and first participant randomised in August 2019, all sites were open by October 2019, and the 9-month internal pilot phase was completed in May 2020. However, the time schedule was impacted by the Covid-19 pandemic. There was a marked reduction in referrals made to CAMHS services during national lockdowns and when

schools were closed. Despite this, we were able to continue recruitment to the trial. We are currently in discussions with the study funder about a proposed time extension. The actual study timelines will be reported in the final study report.

Data management and analysis

Comment 20: Could the authors specify which software will be used for the statistical analyses? In the case of qualitative analysis, they mentioned NVIVO, but did not indicate which ones will be used for quantitative analysis.

Response: A sentence has been added to the statistical analysis section confirming that Stata version 17 (or later) will be used for the statistical analysis (page 19, line 414).

References

Comment 21: The reviewers would appreciate a review of the current literature, especially in relation to the theoretical introduction and the studies conducted. More than 85% of the References, which the manuscript is based on, are older than 5 years and there is only one "current" reference from 2019 (which is two years old). Authors are kindly requested to update the references.

Response: The Introduction has been updated to include more recent studies (page 4, lines 113-116).

Comment 22: A formal revision of the references is suggested. Following the line of the journal, it is recommended to use the Vancouver citation style.

Response: Thank you, we have now revised the references into the Vancouver citation style.

Other comments

Comment 23: The warning "Error! Reference source not found" appears in numerous sections of the manuscript (page 5 line 29; page 8 line 39, page 9 lines 19 and 25; page 10 line 40; page 11 line 48). The authors are kindly requested to check the error.

Response: Apologies, this has now been resolved.

Please note, that in order to address reviewer and editor comments our submitted revised manuscript is now 339 words above the 4000 word guidance.

Thank you for taking the time to read our manuscript. We appreciate all your useful comments.

Yours sincerely,

Prof Kapil Sayal, on behalf of the STADIA team

VERSION 2 – REVIEW

REVIEWER	Ramchandani, Paul University of Cambridge, Education
REVIEW RETURNED	02-Mar-2022
GENERAL COMMENTS	The authors have addressed the issues raised in the previous round. I congratulate the authors on a clearly written protocol. I look forward to reading the results of the trial in due course.

VERSION 2 – AUTHOR RESPONSE

Reviewer 1 comments

Comment: The authors have addressed the issues raised in the previous round. I congratulate the authors on a clearly written protocol. I look forward to reading the results of the trial in due course.

Response: Many thanks for your kind comments and for taking the time to review our manuscript.

Thank you again for your very useful comments.

Yours sincerely,

Prof Kapil Sayal, on behalf of the STADIA team